# The Short-Term Reliability of the Conceptualised ‘Combat Readiness Assessment’

**DOI:** 10.3390/ijerph19116486

**Published:** 2022-05-26

**Authors:** Daniel Moore, Martin Tayler, Stephen Moore

**Affiliations:** 1Occupational Performance Research Group, University of Chichester, Chichester, West Sussex, PO19 6PE, UK; 2Sport and Exercise, Health and Life Sciences, Teesside University, Middlesbrough, TS1 3BX, UK; m.tayler@tees.ac.uk; 3Ministry of Defence, Lyneham, Wiltshire, SN15 4XX, UK; moore969@btinternet.com

**Keywords:** military, soldier, combat, fitness, reliability, fitness-assessment

## Abstract

Military fitness testing has historically assessed individual fitness components. Fitness assessments of this nature do not adequately monitor the physical requirements of military operations. The development of a more combat-specific fitness test would enhance accuracy in assessing the soldiers’ readiness for the demands of war. This study aimed to assess the short-term reliability of the conceptualised ‘Combat Readiness Assessment’ (CRA) following a single familiarisation trial with 21 male phase-two British Army Royal Electrical and Mechanical Engineer (REME) recruits (age (years) 19.7 ± 2.5) split into two groups (N = 11 and 10) to conform with recruit availability. The CRA was designed to be a multifaceted fitness assessment aimed at replicating the physical demands of a combat situation for military personnel. **Methods:** Three repeated assessments of the CRA were completed over a 10-day period (trial one as a familiarisation) to assess the short-term reliability of the CRA post familiarisation. The CRA was completed carrying a 4 kg rifle (SA80 A2) and involved a 400 m (M) run wearing an 11 kg backpack (removed after the 400 M) followed by weighted carries, sprints, casualty drags and agility tasks. Completion time (seconds) was recorded to assess performance. Intraclass correlations (ICCs) (2,1) with 95% confidence intervals (CI), the standard error of the measurement (SEM) and the coefficient of variation (CV) were calculated for completion time for trials 1-3 (T1-3) and 2-3 (T2-3) to assess reliability post-familiarisation. Mean Absolute Percentage Error (MAPE) was calculated for T1-2 and T2-3. Descriptive statistics were calculated for completion time for T1-3 and T2-3. **Results:** The reliability following a familiarisation trial from T1-3 (ICC: 0.75; SEM: 7.1 s; CV: 9.97%) to T2-3 (ICC: 0.88; SEM: 6.4 s; CV: 10.05%) increased. Mean trial time decreased post familiarisation from T1-3 (210.9 ± 21.03 s) to T2-3 (206.3 ± 20.73 s). **Conclusions:** These findings are inconclusive regarding the short-term reliability of the CRA. The small sample size resulted in wide 95% confidence intervals associated with the ICCs, making the true ICC value hard to determine. The ICCs and MAPE suggest that the reliability of the CRA increased following a familiarisation trial, but this requires further research with a larger cohort to determine with confidence.

## 1. Introduction

Across all armed forces, there is an expectation that military personnel should be of a required fitness standard to meet the demands of war. The various actions typically evidenced in combat situations include jumping, climbing, pushing, sprinting, crawling, bounding, rolling, stopping and carrying heavy loads long distances [1]. Military fitness testing has historically assessed individual fitness components [2,3,4,5,6]. Fitness assessments of this nature do not adequately monitor the physical requirements of military operations [7,8]. Numerous authors [6,9] argue that the development of more combat-specific fitness assessments would enhance accuracy in assessing the soldiers’ readiness for the demands of war.

Few studies document the development of various batteries of tests designed to test all components of ‘combat fitness’ [10,11]. The US Army developed a pre-employment fitness assessment called the ‘Occupational Physical Assessment Test’ (OPAT) [10] to assess military-specific fitness standards of potential soldiers prior to joining, and it is subsequently followed by the ‘criterion measure task simulations’ (CMTS) assessment at the end of basic training. The developed testing protocol utilises task simulations to predict the performance of specific military tasks. A development and reliability study involving 751 new recruits utilising 32 different task simulations was completed during assessment development. The final OPAT test battery developed includes four measures; standing long jump, seated medicine ball put, deadlift and the beep test. These four measures were shown to correctly indicate pass/fail status at the end of training CMTS performance in 76% of the new recruits tested [10].

Rohde et al. [11] conceptualised a pre-deployment assessment of basic military fitness in the German Armed Forces called the ‘Basic Military Fitness Tool’ (BMFT). The BMFT consists of a single timed assessment with four distinct phases, all designed to replicate combat-specific scenarios. Additionally, Rohde et al. [11] only reported one trial of the BMFT with 122 soldiers following a familiarisation trial, so little is known about the reliability of the assessment post familiarisation.

Assessments such as the OPAT and CMTS utilise assessments to test individual fitness components that were identified as essential to combat and operational demands of soldiers. These assessments utilised thorough statistical analyses to further understand the validity and reliability of the assessments. Despite the strength of the statistical analysis in developing such assessments, they do not replicate the potential varying physical demands of a combat scenario that does not have breaks between demands being placed on different/multiple fitness components [11,12]. The development of an ecologically valid and reliable combat-specific military fitness assessment that does assess these demands is required and has the potential to change how both annual and pre-deployment fitness tests are conducted across Armed Forces globally.

The current study was designed to assess the short-term reliability of the conceptualised Combat Readiness Assessment (CRA). The assessment was developed from discussions with the Ministry of Defence (MOD) and British Army Physical Training Instructors (PTIs) and researchers. The aim was to develop a multifaceted combat-specific fitness assessment that was easy to administer across groups of soldiers to assess the potential fitness components required in a wartime operational scenario. A quick-to-administer assessment would also be beneficial in assessing groups of military personnel quickly where time constraints exist.

## 2. Materials and Methods

The CRA was devised by combining prior research from the BMFT [11] and expert input from current Military and MOD PTIs and academic researchers. The assessment was designed to be a multifaceted fitness assessment aimed at replicating the physical demands of a combat situation for military personnel who may encounter a combat scenario. Phase 2 REME recruits were available as suitable participants to complete this reliability study. The researchers did not have access to additional military cohorts for this research. The assessment layout is displayed in Figure 1.

All participants were male phase-two British Army Royal Electrical and Mechanical Engineer (REME) recruits. These recruits had advanced from Phase One Basic Training, which is a 14-week course when they first joined the British Army, and were at different stages of a Phase Two REME-specific training course.

Anthropometric measurements of all participants were recorded three days before the first CRA trial. The CRA testing period, with repeated assessments over a 10-day period, was designed to assess the short-term reliability of the assessment following a familiarisation trial whilst being a short time period to ensure recruit fitness levels remained stable for all of the testing period. All participants completed the CRA three times during this period, with the first trial acting as a familiarisation trial. Trials two and three were performed at the same time of day to facilitate the effects of circadian rhythm affecting task performance [13]. The familiarisation trials were completed at a different time of day to adhere to recruit availability.

A minimum of three days was afforded as recovery between trials, ensuring fitness levels did not change during this period and avoiding fatigue during any of the trials. Recruits were advised not to perform strenuous exercise in the three days before their first assessment. The proposed measure for the CRA is completion time. The completion time (seconds) was recorded for each recruit in all completed trials.

### 2.1. Participants

The data was collected at Ministry of Defence (MOD), Lyneham, with male phase-two British Army REME recruits aged 18–26 (years) utilised as participants for the study (*n =* 21; age: 19.7 ± 2.5; height: 177.86 ± 6.5 cm; body mass: 76.95 ± 10.2 kg: body fat: 11.9 ± 2.56%). The 21 subjects were organised into two groups (troops); group A (*n* = 10; age: 20.70 ± 2.7; height: 180.90 ± 3.8 cm; body mass: 81.10 ± 7.2 kg; body fat: 12.09 ± 2.4%) and group B (*n* = 11; age: 19.0 ± 2.2; height: 174.45 ± 7.0 cm; body mass: 73.72 ± 11.8 kg; body fat: 12.05 ± 3.0%) to adhere to their availability during training. Skinfold measures were perfomed by one researcher; intrarater reliability was not calculated for this measure.

The sample size was determined by the number of recruits available during the time research access was afforded at MOD Lyneham. Sample size and power calculations were not calculated due to the small sample size available. The recruits were assessed in two separate groups to align with their physical training sessions and availability.

All participants received both verbal instructions and participant information sheets about the nature of the study. Thereafter, they completed a medical questionnaire and informed consent forms, hereby consenting to data collection and providing basic information. Ethical approval was granted by the School of Social Sciences, Humanities and Law Ethics Committee at Teesside University.

### 2.2. Procedures

Anthropometric data collection was completed three days prior to completing the first CRA trial. At this stage, all participants were instructed not to complete any strenuous exercise 24 h prior to any CRA trial. Anthropometric data utilised included height (cm) (Seca, Leicester Portable Height Measure, Seca Weighing and Measuring Systems, Birmingham, England), body mass (kg) (Seca, 869, Seca Weighing and Measuring Systems, Birmingham, England) and body fat percentage (BF%) (Harpenden skinfold calliper, Burgess Hill, West Sussex, UK) measuring skinfold thickness at four sites (biceps, triceps, subscapular and supra-illiac), the full protocol can be found in Durnin and Womersley (1974) [14].

The familiarisation trial (T1), T2 and T3 were all conducted with the same protocols. Before every CRA, the standardised warm-up was completed as a troop between two lines 20-metres (m) apart with one effort defined as a 20 M distance. The warm-up was standardised across all trials and consisted of gradual speed development and dynamic stretching sequence in the following order; walk X2, light jog X4, half pace run X4, knee raises X2, heel flicks X2, walking lunges with hands touching the floor X2, backward running X2, side steps X4, ¾ pace run followed by a walk X3, sprint followed by a walk X3.

The CRA layout is shown in Figure 1. The assessment was completed in field boots, combat trousers, a combat shirt and carrying a 4 kg rifle (SA80 A2). The initial 400 M run was completed wearing an 11 kg backpack (bergen). The order each CRA trial was completed in was as follows: a 400 M run around the outside of the course with the backpack being put down and the rifle being placed into the slung position (behind back) upon completion of the 400 M run. The next phases consisted of a 10 M sprint to two 20 kg jerry cans (petrol cans), a 90 M (3 × 30 M turning at the line) 20 kg jerry cans carry (one in each hand), a 10 M sprint to a 50 kg casualty dummy, a 20 M backward 50 kg casualty drag, a 10 M sprint to the agility zone, completion of the agility zone box (shown in Figure 2) and a 60 m sprint to the finish line.

All CRA trials were completed during the recruit’s programmed physical training (PT) sessions. The same Army PTI conducted all standardised warm-ups and assessment instructions and started each assessment. Timekeeping responsibilities were shared between the PTI and the researcher. All CRAs were completed on the same marked out flat 400 M concreted location as illustrated in Figure 1. Only the middle three lanes were used; the participants used the same lane for each assessment to ensure standardisation for each CRA trial completed. The temperature ranged between 16 and 21 °C, and conditions were either dry or very light rain across all trials.

During each CRA trial, the completion time was recorded to the nearest second using a digital stopwatch.

### 2.3. Statistical Analyses

After confirming normal distribution, independent *t*-tests were performed between all anthropometric variables (height, body mass, BF% and age) for groups A and B to ascertain any statistically significant group differences. Descriptive statistics were reported as mean ± SD for both groups and groups ‘A & B’ separately to allow for comparison of groups.

Descriptive statistics were calculated for completion time and reported as mean ± SD. The normal distribution of completion time data was confirmed using descriptive methods (skewness, outliers and distribution plots) and inferential statistics (Shapiro–Wilk test). Intraclass correlations (ICCs) (2, 1) with 95% confidence intervals (CI) were calculated for completion time using a two-way fixed-effects model, ensuring both systematic and random errors were considered [15,16]. ICCs were used to classify reliability as values less than 0.5, between 0.5 and 0.75, between 0.75 and 0.9 and greater than 0.90 are indicative of poor, moderate, good and excellent reliability, respectively [17].

As an index of absolute reliability, the standard error of the measurement (SEM) was calculated [18,19]. The SEM was determined as SEM = √MSE. This ensured that: (1) the SEM was not by the between-participant variability (as the ICC is); (2) the SEM was calculated separately from the ICC, and (3) only random error was considered. 

The coefficient of variation (CV) was calculated (CV = (SD/mean) 100). The CV conveys the variability of a ‘score’ as a percentage in relation to the group mean.

The ICC, SEM and CV were determined for completion time for trials 1-3. In order to evaluate the impact of a single familiarisation trial on reliability, these same processes were completed for trials 2-3, excluding the first trial from the analysis.

In addition to the ICCs, Mean Absolute Percentage Error (MAPE) was calculated for T1-2 and T2-3 to assess the difference in the error in the measurement from familiarisation-T2 and T2-T3.

Statistical analysis was performed using the statistical package R Studio Team, RStudio Version 1.0.153-© 2009-2017, (Boston, MA, USA), and manual calculations were performed using Microsoft Excel Version 1910, (Redmond, WA, USA). For all statistical tests, alpha was set at 0.05. Descriptive statistics are presented as mean ± SD.

## 3. Results

The recruit height between groups A and B was statistically significantly different (t (19) = 2.583, *p* = 0.018). Age, weight and BF% were not statistically significantly different between groups (*p* > 0.05). Anthropometric mean values (±SD) are shown in Table 1

Mean trial values and the corresponding reliability outcomes for completion time are shown in Table 2. The completion time ICC classification for trials 1-3 was moderate with 95% CI ranging from poor to excellent. For trials 2-3, the completion time ICC classification was good with 95% CI ranging from moderate to excellent. MAPE from T1-2 to T2-3 was 5.79% to 3.77%, respectively.

Individual completion times across all three trials are shown in Figure 3.

## 4. Discussion

An aim during the development of the CRA was to create a multifaceted combat-specific fitness assessment to assess the potential fitness components required in a wartime operational scenario. Although the analysis utilised does not explore the individual contributions of different fitness components and how they contribute to the completion time, it is important to discuss the fitness components utilised. Typically combat scenarios include tasks that require high levels of anaerobic fitness, such as sprinting, rushes, direction changes, drags and carries [20]. All the actions utilised in the CRA are actions representative of tasks similar to those described. 

The CRA, however, does not solely rely on anaerobic fitness. Muscular strength is an essential fitness component within the CRA; strength is required to perform casualty drags, jerry can carries, load carriage and sprinting [21]. Furthermore, due to the duration of the CRA, aerobic fitness is an essential component. Adequate adenosine triphosphate (ATP) levels are not sustainable for the duration of the CRA. Thereby, oxygen is required to assist in ATP regeneration alongside fueling energy demands via the aerobic energy system [22].

The CRA is a conceptualised combat-specific assessment, and therefore no previous research has been completed on its reliability. Following a review of the military fitness assessment literature, the BMFT [11] is the only combat-specific military assessment that shares commonalities in the assessment design with the CRA. One of the limitations of the BMFT is an absence of a reliability study, thereby making it difficult to compare the CRA to the BMFT. Therefore, explanations and discussion include comparisons to the reliability of pre-existing military fitness assessments with varying protocols and what the results may mean to the application and reliability of the CRA. 

The data presented provide a detailed reliability assessment because it was examined across repeated trials, which are required to accurately assess habituation [23]. The best-planned reliability studies are designed to have multiple retests [23]. Through repeated tests performed on male phase-two REME recruits, the aim of this study was to complete a comprehensive evaluation of the short-term reliability of the CRA. The proposed measurement practitioners would utilise to assess performance in the CRA would be completion time. An increased ICC value from trials 1-3 to 2-3 from 0.75 to 0.88, respectively, indicates that by following a single familiarisation trial, the CRA completion time reliability increased from moderate to good [17].

When conveying reliability data, it is important that a distinction is made between absolute and relative reliability [24]. Absolute reliability is represented by the CV [23], while correlation coefficients are often used to assess relative reliability [6]. In this study, relative reliability was evidenced in the ICCs. Confidence intervals should be reported to identify the likely range of values that contain the true ICC to the population [6]. The 95% confidence intervals utilised in this study identify the range of the true ICC values to the population and what classification band/s the range sits within.

The confidence intervals in both T1-3 and T2-3 are relatively wide (poor to good and poor to excellent, respectively), identifying that whilst the ICCs appear to indicate increased reliability following a familiarisation trial, there is uncertainty in the measure representing the true ICC to the population. Furthermore, T1-3 and T2-3 have a similar CV (9.97% and 10.05%, respectively); this represents quite a large variability from the mean with respect to absolute reliability. The SEM decreased in T2-3 compared to T1-3, which suggests following T1, the test–retest reliability increases. Additionally, a decrease in the MAPE from 5.79% (T1-2) to 3.77% (T2-3) may suggest that the CRA measurement of completion time is more reliable following a familiarisation trial. Thereby, it is assumed that the wide confidence intervals are due to a low sample size rather than an error in the measurement. In order to gain more certainty about the true ICCs and CVs, a larger sample size based on a power analysis should be used in the future.

It is likely that the decrease in CRA times following T1 is due to habituation to the assessment [19]. Hereby, a learning effect has likely taken place where the recruits learned how to complete the CRA more efficiently, which was evidenced in many fitness assessments with athletes [25]. If participants become familiarised with the CRA following one trial, it would be beneficial as an assessment tool that can become reliable following just one trial. However, based on this study alone is difficult to conclude this with certainty.

The ICCs alone indicate that by following a familiarisation trial, reliability increases from moderate to good; however, the wide confidence intervals CV and SEM values identify that further research with a larger cohort is required to better understand the short-term reliability of the CRA. Additionally, the completion times appear to still be decreasing from T2-3, which may indicate that habituation is not complete after one CRA trial, and the learning effect is still occurring.

While the results of this study indicate that the CRA completion time may be a reliable short-term assessment following a familiarisation trial, this may only be indicative of this small population of male REME recruits. The subjects were determined by recruit availability at the time of data collection and did not include any females. Previous research surrounding load-carriage tasks suggests that there is a significant sex difference in anaerobic-task performance during load-carriage [12] alongside significant sex differences in carrying position during load-carriage tasks [10].

Further research with male and female participants in a substantially larger cohort would be beneficial to help assess the CRAs’ reliability with both sexes. Additionally, the weight of all equipment, such as the 11 kg backpack, 20 kg jerry cans and 50 kg dummy used in the CRA, is not reflective of all combat scenarios, and the environmental requirements during a combat situation could vary significantly [7]. However, a combat situation is a dynamically changing environment; thereby, it is impossible for any assessment to be ecologically valid for all combat scenarios [11].

Ideally, during this study, the CRA familiarisation would have been completed at the same time of day as the following two trials to allow for the effects of the circadian rhythm. Unfortunately, due to recruit availability and their pre-determined physical training times, this was not possible. Circadian rhythm has been linked to differences in neurobehavioral variables, including fatigue, boredom and distractibility [9]. Crucially, however, the two CRA assessments for each of the two groups following the single familiarisation trial were at the same time, thereby allowing for the effect of circadian rhythm post familiarisation.

In future research, a similar study design could be utilised with a larger cohort guided by a sample size power calculation to ensure that the reliability data obtained produce more certainty around the relative and absolute reliability of the CRA. In addition to testing the reliability, it would be advantageous to assess the validity of the CRA in future research. Additional measures such as including intra- and inter-rater reliability measures of the researchers’ skinfold measurements would enhance any future research into the CRA. Furthermore, dietary intake was not monitored. An understanding of the recruits’ nutritional intake prior to all CRA trials would be beneficial in the next phase of CRA research.

## 5. Conclusions

The CRA is still a conceptualised assessment that has not been introduced across a branch of any Armed Forces. This study formed part of the CRA development with phase-two REME recruits. However, the results of this research are somewhat inconclusive as to the reliability of the CRA. 

The ICC differences alone from T1-3 compared with T2-3 may suggest that if the CRA was implemented in a branch of the Armed Forces, completion time might be a reliable short-term assessment method to determine performance following a familiarisation trial. In addition to this, the MAPE reduction from 5.79% (T1-2) to 3.77% (T2-3) may suggest that the CRA measurement of completion time is more reliable following a familiarisation trial and that the error is not in the measurement.

The CIs associated with the ICCs (T1-3 and T2-3) are wide with high CVs and only a small change in the SEM. Hereby, it must be concluded that the CRA requires further research to fully understand the short-term reliability of the assessment. Additional research with a far larger cohort is required to accurately assess the short-term reliability of the CRA. If the increase in ICC from moderate to good was matched or further enhanced with narrower CIs and a similar decrease in the MAPE in a larger study, there would be far greater evidence of the short-term reliability of the CRA. Such research should include a wide demographic of military personnel to substantiate the CRA’s short-term reliability.

## Figures and Tables

**Figure 1 ijerph-19-06486-f001:**
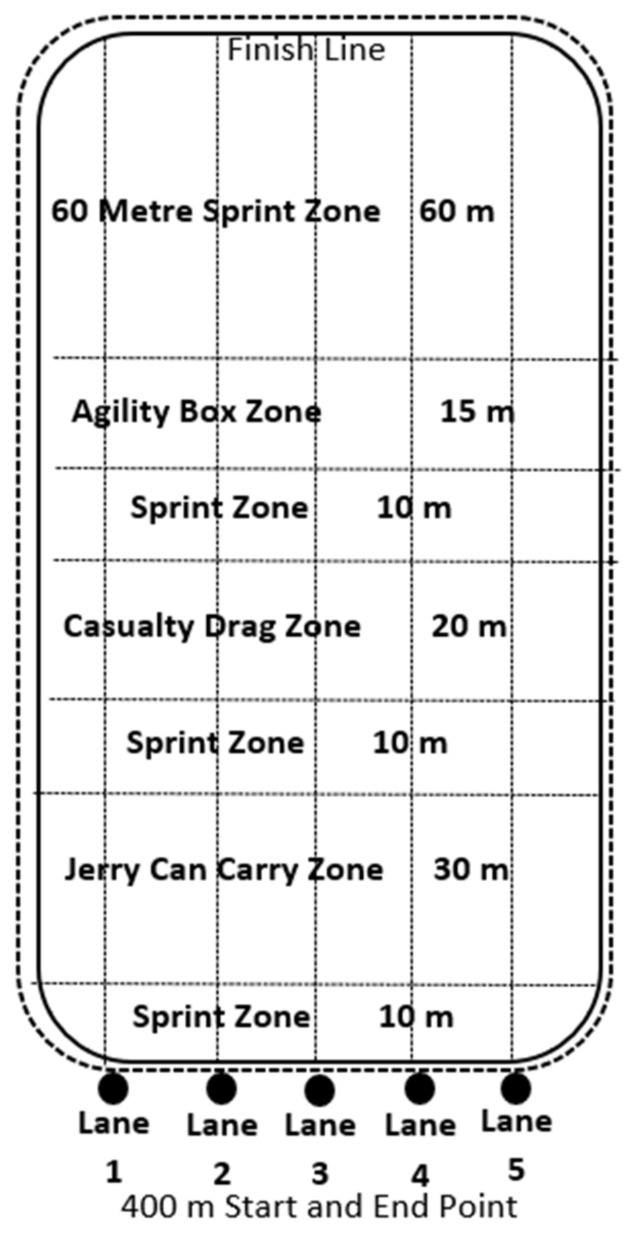
Combat Readiness Assessment Layout. Showing lanes 1–5 and each zone of the assessment.

**Figure 2 ijerph-19-06486-f002:**
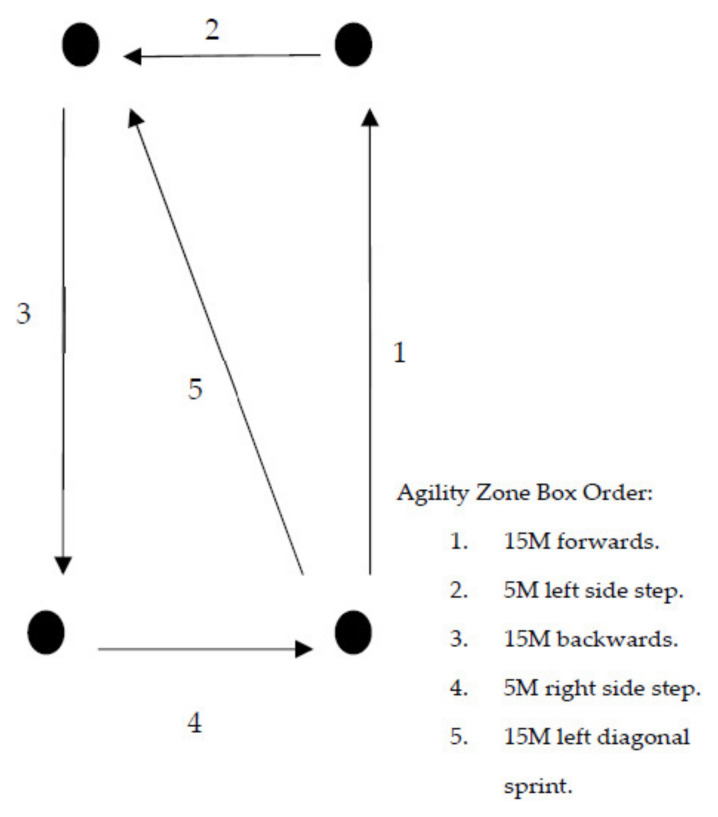
Agility Zone Box Protocol.

**Figure 3 ijerph-19-06486-f003:**
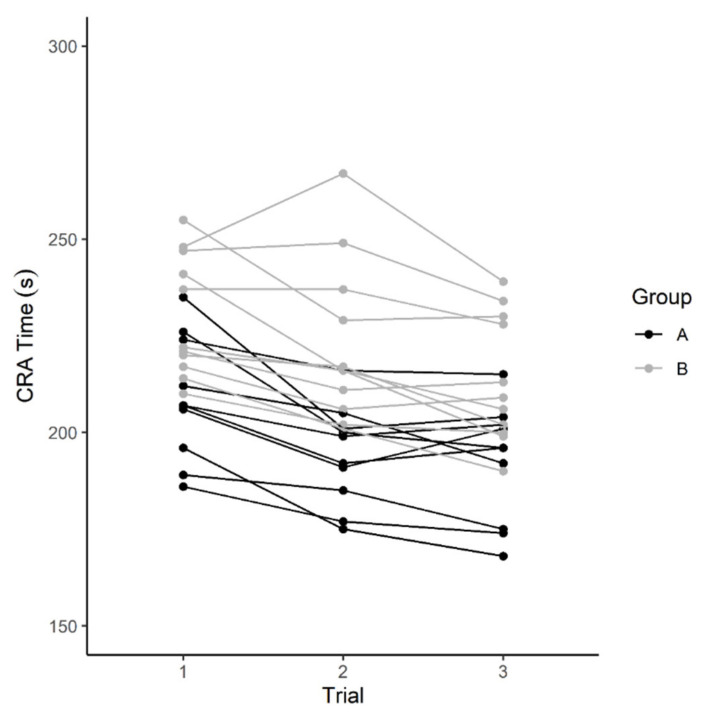
Individual changes in completion time across all three CRA trials.

**Table 1 ijerph-19-06486-t001:** Mean values for recruit age and anthropometric data.

	Age	Height (cm)	Body Mass (kg)	Body Fat %
**Groups A and B** **(N = 21)**	19.7 (±2.49)	177.86 (±6.51)	76.95 (±10.2)	11.9 (±2.56)
**Group A** **(N = 10)**	20.7 (±2.83)	180.9 (±4.04)	81.1 (±7.01)	12.1 (±2.05)
**Group B** **(N = 11)**	19 (±2.19)	174.45 (±6.99)	73.73 (±11.82)	12.05 (±3.02)

**Table 2 ijerph-19-06486-t002:** Mean trial values and the corresponding reliability outcomes for CRA completion time from trials 1-3 to trials 2-3.

	T1-3	T2-3
Completion time (seconds)	210.9 (±21.03)	206.3 (±20.73)
ICC (95% CI)	0.75 (0.32–0.91)	0.88 (0.64–0.95)
SEM (seconds)	7.1	6.4
CV (%)	9.97	10.05

Abreviations in Table 2: Intraclass Correlation = ICC; Confidence Interval = CI; Standard Error of the Measurement = SEM; The Coefficient of Variation = CV.

## Data Availability

The datasets used and analysed during the current study are available from the corresponding author on reasonable request.

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
