# Peer review of "The Short-Term Reliability of the Conceptualised ‘Combat Readiness Assessment’"

_ijerph, 2022, doi:10.3390/ijerph19116486_

Round 1

Reviewer 1 Report

This study suggested very interesting results and experimental suggestions. 
Also, this paper is well written with logical flow. 
However, I found some minor limitations and suggest revision related these issues. 
This paper deserves to be published after minor revisions.

1. Check the references. Read the instruction to authors in detail.
2. English language and style are fine/minor spell check required.
3. Please add to the abstract that the study was conducted in two groups of 21 male soldiers.
4. Page 3: I would like to add information about the target group in a table. It seems to be necessary to confirm that there is no difference in general information between the two groups. If there is no difference between the two groups, the comparison of the results will be meaningful.
5. Page 5: For the abbreviations in Table 1, write the full name below the table.

Author Response

Dear Reviewer,

Thank you for your feedback on the manuscript.

We are happy to make the revisions you have suggested.

  • The references check and spell check have been performed.
  • The 2 groups is now highlighted in the abstract.
  • A table has been added showing the age and anthropometric characteristics makeup of the two groups utilised in the study alongside the sample as a whole.
  • The full name of the abbreviations has been written below table 1 (now table 2).

Kind regards,

Daniel Moore

Reviewer 2 Report

Overall, this was a well-written manuscript, below are my comments

Introduction

  1. Lines 39-40- You state few studies, but don't cite any
  2. Line 48- Do you mean beep test?
  3. Lines 58-60- what do you mean?
  4. line 62- short time span?

Methods

  1. When you talk about performing the trials at the same time of day to account for circadian rhythm you should cite Pilcher's work on this.
  2. How did you determine sample size? Was there an priori power analysis? Did you perform a power analysis during the trial?
  3. Line 106 and also later near lines 251- Please provide ethics approval number
  4. How many researchers performed the skinfold assessment? What was the inter and intra-rater reliability
  5. I'm confused about groups A and B. What is the point of having groups in the analysis since both groups performed the intervention. Other than describing the fact that they were split into 2 groups that performed the CRA at 2 different times, I don't see the relavance of doing that.
  6. I can understand that you're using the ICC between trials 1 and 3 to illustrate how reliability improves from 2-3, but I think you miss the point. With 1 simply being familiarization, most of the data in this work suggests that there will be significant improvements between T1 and T2 (as seen in your results). However, from what i can see time still continued to decrease between T2 and T3. Could this have been a continuation of the learning effect?
  7. When testing reliability of 2 quantitative measures you should use the Bland Altman and Mean Absolute Percent Error

Results

I don't know if I'd state that your 95% CI indicates near perfect reliability. The width of your 95% CI did decrease when comparing T1-T3 and T2-T3, suggesting that your measurements got closer to each other, but it doesn't indicate that the reliability was near perfect. The 95% CI is the range that can contain your correlation coefficient.

Discussion

I feel that the authors have extrapolated too much in the discussion. I would strong suggest the researchers run the Bland-Altman and the MAPE and write their results accordingly. Those will tell you significantly more information. 

Author Response

Dear Reviewer,

Thankyou for your feedback on this manuscript.

I believe I have address the points noted in the introduction in the revised submission.
1. References inserted.
2. Corrected to Beep test 
3. more detail added.
4. reworded to make the point clearer.

Methods

1. Cited one of Pilchers studies which supports the point regarding circadian rhythm
2. Sample size was only determined by recruit availability in the short time period we had. This and a lack of priori power analysis has now been mentioned then further discussed as a limitation of the study and recommendation for future research on the CRA.
3. Ethics reference number now provided
4. 1 researcher performed skinfold. No intraclass correlation was calculated. added to limitations
5. This has been further clarified in text. It was solely to fit in around the recruits availability and their physical training timings.
6. The learning effect and the potential of its continuation into T3 has now been added to the discussion.
7. We have utilised ICCs with 95% CIs, CV and SEM and believe that this gives a strong and appropriate statistical reliability analysis for this research. I do understand that Bland Altman can also be used for such analysis but we have followed literature supporting the use of ICCs for such analysis (ref below this point). I do also understand that there is literature supporting the use of Bland Altman.

I have however amended the ICC classification names and boundaries to a more conservative recommendation than the previous submission to avoid overselling the strength of the reliability (line 212). Further attention has also been paid to the CIs, CVs and SEM.

Koo, T. K., & Li, M. Y. (2016). A Guideline of Selecting and Reporting Intraclass Correlation Coefficients for Reliability Research. Journal of chiropractic medicine15(2), 155–163. https://doi.org/10.1016/j.jcm.2016.02.012

Results

As above we have changed the classification names and boundaries to ensure that we are not overselling the strength of the reliability.
Additionally added discussion regarding the 95% CIs, CVs and SEMs has been detailed later in the discussion.
I believe the additional explanations/discussion of the 95% CIs, CVs and SEM followed in the discussion help address this point.

Discussion

I can see the point you are making and have tried to address this with further analysis of the additional reliability measures beyond the ICCs. 
The discussion is more focused on the results being inconclusive due to small sample size and recommends future research with a larger cohort based on a power calculation.

Kind Regards,

Daniel Moore

Reviewer 3 Report

In this work, Dr. Moore and colleagues conducted a reliability study on the components of "combat-fitness" in 21 young male subjects of the British Army. The conceptualised program, namely ‘Combat Readiness Assessment’, resulted interesting and an ecological one. The methodology was reasonable and the design sound.

If the battery may result easy-applicable and valid as compared to other test batteries, what makes me perplexed is the "claimed" increasing reliability levels along the the three trials. In fact, this is expected and known as a "learning effect" and therefore I wonder why the Authors reported that as positive achievement of the study. If I am misinterpreting, please may the Authors clarify this aspect.

  • In addition to that, on a singular note, was the familiarization trial properly controlled ? How ? Please may the Authors specify this.

  • How did the Authors control the participants' fitness level throughout the trails so to maintain it stable?

Author Response

Dear Reviewer,

Thankyou for your feedback on the manuscript.

Further discussion around the 95% CIs and CVs and SEMs alongside the ICCs has now been included which changes the previously claimed reliability from the last submission. Learning effect has now been discussed and I believe it addresses your concern.

Additional note (line 137) has been added that all trials including the familiarisation were conducted exactly the same. I hope this addresses your question regarding controlling T1.

Additional note (lines 98/99) has been added to clarify the stability of the recruits fitness levels.

Kind regards,

Daniel Moore

Reviewer 4 Report

Dear Authors,
We were pleased to review and comment to the authors on the manuscript titled "The Short-Term Reliability of the Conceptualised 'Combat Readiness Assessment'.
The manuscript is well written and organized in its structure, it is methodologically well outlined, the data analysis is performed correctly and the discussion is reasonably constructed. However we are afraid that its eventual publication in the IJERPH may have a reduced scientific utterance due to the fact that the topic has, in our opinion, a restricted interest to the scope of researchers and readers linked to the military world. We reported this ours concern to the Editor-in-chief.
However, we have some comments and suggestions that may help to improve the quality of the manuscript that we point out to the authors below:

Introduction Section
#Please clarify the rationale for stating that the existing fitness assessments tests and mentioned by the authors of the not adequate monitor physical requirements of military operations.

#The authors state that numerous studies argue that the development of more combat-specific fitness assessments would enhance accuracy in assessing the soldiers’ readiness for the demands of war. But in the army there are different task assignments with very different requirements. How do the authors support that the test now proposed, the "Combat Readiness Assessment (CRA)" includes all components of 'combat fitness'?

#line 68 please state the meaning of the abbreviation MOD in full, as this is the first time it is mentioned in the text

Materials and Methods Section
# Please clarify the conceptual basis underlying the construction of the "Combat Readiness Assessment (CRA)" test, namely its suitability to assess male soldiers' readiness for the demands of war, in the performance of tasks assigned and required to soldiers in the phase- two British Army Royal Electrical and Mechanical Engineer (REME) recruits.

Participants Section 
# please clarify if the participants are already military or if they are still aspiring military, since it is mentioned that they are recruits (line 97)
# please indicate how British Army Royal Electrical and Mechanical Engineer recruits are recruited and whether they have passed some sort of selection test before becoming recruits in the phase-two British Army Royal Electrical and Mechanical Engineer
# please clarify the rationale for dividing the sample into 2 groups (group A and group B)
# Please indicate the sample size and power calculation. Sample size calculation is to determine the number of participants needed to detect a clinically relevant "treatment effect" and power calculations allow the reader to know how many subjects are required in order to avoid a type I or a type II error.
# please indicate the ethical approval number

Procedures section
 Statistical Analyses
#line 152 - please clarify what the references "(2,1)" mean

BW

João Paulo Brito

Author Response

Dear Reviewer,

Thankyou for your feedback on this manuscript.

Regarding your concern over suitability to the journal. I understand however we believe that due to the focus of this special edition it is suitable for this edition.

Introduction

# I have added some extra wording regarding the existing fitness tests and what they may be lacking in lines 66/67 i hope this helps address this.

# Discussion around the CRA and the difficulties around creating a fitness assessment for all combat scenarios is detailed more in the discussion in the paragraph at lines 278 - 284

#Full Ministry of Defence (MOD) now written

Materials and Methods

# additional wording regarding the CRA targeted groups and why REME recruits were used has been added (lines 84 - 87)

#Further detail explaining that they have passed phase 1 Army training (14-week course) has been included (lines 92 - 94)

# The recruits were assessed in two separate groups to align with their physical training sessions and availability. (lines 120/121)

#Sample Size is addressed at line 118. Further discussion around this is now throughout the discussion and limitations.

# Ethical Approval reference number now included

Procedures

# (2,1) refers to the type of ICC. (Each subject is measured by each rater, and raters are considered representative of a larger population of similar raters. Reliability calculated from a single measurement).

Kind Regards,

Daniel Moore

Reviewer 5 Report

Dear authors,

First, I would like to congratulate you on completing such an interesting research project given that there are limited data in this area. As far as the manuscript goes, it’s well-written and informative. The authors have been able to touch on the critical points throughout the manuscript which makes it really an enjoyable read. The only question that I have is whether dietary intake ( maybe supplements) has been monitored. If not, simply add as a limitation of this project.

Author Response

Dear Reviewer,

Thankyou for your feedback and kind words regarding our manuscript. 
Dietary intake was not monitored. We are happy to add this as a limitation of the work.

Kind regards, 

Daniel Moore

Reviewer 6 Report

This manuscript entitled “The Short-Term Reliability of the Conceptualised ‘Combat Readiness Assessment’” aimed to develop a multifaceted combat-specific fitness assessment that was easy to administer across groups of soldiers to assess the potential fitness components required in a wartime operational scenario. 
There are some concerns about the study that preclude the recommendation for publication. 

Why individual fitness components assessments are not adequate to monitor the physical requirements of military operations? 
Would you please clarify the gap in the literature and what this study adds?

Reliability is an important point to having a good test battery, but there are other measures as important as reliability. For example, validity. 
The sample is too small. This is one of the biggest limitations of this study. Can authors provide sample calculations for this kind of Analysis? 
From the abstract, it is not possible to understand what is Combat Readiness Assessment’
Is the quality of the performance of the exercise assessed in this assessment? I assume it is important. It should be discussed. 
There is not sufficient information to replicate this protocol in another context.
Why the 21 subjects were organized into two groups? In the Statistical Analyses section, Independent t-tests were performed to compare groups A and B. Why? Probably a non-parametric Statistical Analysis fits better here.
The authors may want to discuss in more detail the applied implications of the findings.
The conclusion is not aligned with the results or the main purpose of this study.  

Author Response

Dear Reviewer,

Thankyou for your feedback on this manuscript.

  • I have added some writing in lines 66/67 and 77-79 which aim to give some clarity as to where individual assessments may fall short and the benefit/gap in literature an assessment like the CRA could fill.

  • I agree that the sample size is lacking. We only had limited access and time with the recruits and used what was available. I have addressed how the sample was recruited and the lack of power power calculations in lines 118 - 120. Additionally the with some other changes in 95% CI analysis the sample size limitations and lack of prior size calculations is discussed more thoroughly later in the paper.

  • Lines 15 -17 now provide additional information in the abstract to highlight what the CRA is

  • The reason for two groups is now detailed in lines 120/121

  • A t-test was used as the data fitted the assumptions for a t-test

  • A lot of changes have been made in the discussion and conclusion following more analysis of the 95% CIs, CVs and SEMs. I believe that this addresses more discussion around the implications and aligns with the aims

Kind regards,

Daniel Moore

Round 2

Reviewer 2 Report

I'd like to thank the authors for taking the time to address my concerns. I still have significant concerns about using just the ICCs in your analysis. I think there is some validity to using ICCs however, the ICC does not give you the opportunity to evaluate the bias between the mean differences and estimate an agreement interval. 

The MAPE and the ICC combined will help you determine whether the confidence interval for the ICC is due to low sample size or error in actual measurement. Combining those two pieces of information will give you an opportunity to determine whether you were just lacking power or whether there were just errors in the measurement. The information that you presented does not give me the reader enough information to determine whether I should bother replicating your experiment or not. 

A smaller, but very addressable issue with this MS is I don't know if the data of group A and B were normally distributed or not. If they're not then you can't use a T-test. 

Author Response

Dear Reviewer,

Thanks again for further feedback and advice regarding this manuscript.

This makes sense and I can certainly see the benefit of including the MAPE alongside the current ICC analysis. I have calculated and included the MAPE for T1-2 and T2-3 for the CRA completion times. 
Further edits have been made to the discussion, conclusion and abstract to include the addition of the MAPE to the study and what it means.

Regarding the t-test. That's my mistake I forgot to write that the t-test was completed following confirming normal distribution and have now added that to the sentence in the metods.

Kind regards,

Daniel Moore

Reviewer 6 Report

The major issues in this paper were addressed in the revision. I don't have further questions. 

Author Response

Thanks again for all feedback given.